



# High resolution aerosol concentration data from the Greenland NorthGRIP and NEEM deep ice cores

Tobias Erhardt[1, 2], Matthias Bigler[1], Urs Federer[1], Gideon Gfeller[1], Daiana Leuenberger[1],
Olivia Stowasser[1], Regine Röthlisberger[1], Simon Schüpbach[1], Urs Ruth[2, 3], Birthe Twarloh[2],
Anna Wegner[2], Kumiko Goto-Azuma[4, 5], Takayuki Kuramoto[4, 6], Helle A. Kjær[7], Paul T. Vallelonga[7],
Marie-Louise Siggaard-Andersen[8], Margareta E. Hansson[9], Ailsa K. Benton[10, 11], Louise G. Fleet[10],
Rob Mulvaney[10], Elizabeth R. Thomas[10], Nerilie Abram[12], Thomas F. Stocker[1], and Hubertus Fischer[1]

[1]Climate and Environmental Physics, Physics Institute, and Oeschger Center for Climate Change Research, University of Bern, Bern, Switzerland
[2]Alfred-Wegener-Institute Helmholtz Center for Polar and Marine Science, Bremerhaven, Germany
[3]Robert-Bosch-GmbH, Corporate Research, Stuttgart, Germany
[4]National Institute of Polar Research, Research Organization of Information and Systems, Tokyo, Japan
[5]Department of Polar Science, Graduate University for Advanced Studies (SOKENDAI), Tokyo, Japan
[6]Department of Human Development, School of Humanities and Culture, Tokai University, Hiratsuka, Japan
[7]Section for the Physics of Ice, Climate and Earth, The Niels Bohr Institute, University of Copenhagen, Copenhagen, Denmark
[8]Section for Geogenetics, Globe Institute, University of Copenhagen, Copenhagen, Denmark
[9]Department of Physical Geography, University of Stockholm, Stockholm, Sweden
[10]British Antarctic Survey, Cambridge, UK
[11]Department for Environment, Food & Rural Affairs, UK Government, UK
[12]Research School of Earth Sciences and ARC Centre of Excellence for Climate Extremes, Australian National University, Canberra, Australia

**Correspondence:** Tobias Erhardt (tobias.erhardt@climate.unibe.ch)

**Abstract.** Records of chemical impurities from ice cores enable us to reconstruct the past deposition of aerosols onto the polar ice sheets and alpine glaciers. Through that, they allow us to gain insight into changes of the source, transport and deposition processes that ultimately determine the deposition flux at the coreing location. However, the low concentrations of the aerosol species in the ice and the resulting high risk of contamination poses a formidable analytical challenge, especially if long,

5 continuous and highly resolved records are needed. Continuous Flow Analysis, CFA, the continuous melting, decontamination and analysis of ice-core samples has mostly overcome this issue and has quickly become the de-facto standard to obtain high-resolution aerosol records from ice cores after its inception at the University of Bern in the mid 90s.

Here we present continuous records of calcium ($Ca^{2+}$), sodium ($Na^+$), ammonium ($NH_4^+$), nitrate ($NO_3^{-1}$) and electrolytic conductivity at 1 mm depth resolution from the NGRIP (North Greenland Ice Core Project) and NEEM (North Greenland

10 Eemian Ice Drilling) ice cores produced by the Bern Continuous Flow Analysis group in the years 2000 to 2011 (Erhardt et al., 2021). Both of the records have previously been used in a number of studies but have never been published in the full 1 mm resolution. Alongside the 1 mm datasets we provide decadal averages, a detailed description of the methods, relevant





references, an assessment of the quality of the data and its usable resolution. Along the way we will also give some historical context on the development of the Bern CFA system.

## 1 Introduction

Proxy records from polar ice cores have allowed us to gain detailed insight into the past climate and its variability. They provide us not only with a reference for our theoretical understanding of the Earth system but also important context for anthropogenic climate change. Among the records from polar ice cores, the most prominent ones are those of past atmospheric gas composition, preserved in the bubbles enclosed in the ice; the isotopic composition of the water, preserved in the ice itself; and the records of past aerosol deposition onto the polar ice sheets, preserved in the ice as wide range of chemical impurities. Together, these proxies contain information about different parts of the Earth system, all recorded in the same archive, allowing for detailed multi-parameter studies of the past climate.

Like all these proxies, the aerosol records come with their own set of challenges not only in terms of their interpretation but also in terms of their measurement in the ice matrix. Often, innovative sampling and measurement techniques are necessary to obtain the required analytical precision and resolution. Especially the latter is important for aerosol records as the concentration of impurities in the ice varies on the seasonal time scales, translating into cm to mm-scale variability along the depth in the ice-core record. Together with their low concentrations in the ice and the resulting contamination risks and the requirement for high analytical sensitivity, this poses a formidable challenge for the measurement of continuous records in the often kilometer-long ice cores.

To overcome this challenge was one of the motivating factors to develop the continuous melting and measurement technique, Continuous Flow Analysis (CFA) for ice core samples first described by Sigg et al. (1994). The key concept of the method lies in the way the ice itself is sampled: For CFA, a longitudinal section of ice core is melted vertically, separating the meltwater from the inner part of the ice from that of the possibly contaminated outer part. The meltwater from the inner part is then used for analysis by continuous detection methods or for discrete aliquots for later analysis. In essence, this concept is shared by all CFA systems that have been build since then and remains largely unchanged until today. In the meantime this kind of meltwater sampling has become the gold standard for ice core chemical studies, as it is –except at core breaks– completely contamination free and reduces the tedious and painful manual decontamination in the cold lab to a minimum. This has opened the door for continuous, high-resolution ice core chemistry studies previously deemed infeasible or impossible. Initially, the method was only applied for a limited range of analytes (Sigg et al., 1994) however it was slowly extended (Röthlisberger et al., 2000; Kaufmann et al., 2008) and even includes methods for trace- and ultra-trace elemental concentration measurements (e.g. McConnell et al., 2002; Knüsel et al., 2003; Erhardt et al., 2019a). Furthermore, in solid ice, the sampling can even be performed air-tight such that the air enclosed in the ice can be extracted from the meltwater stream. Accordingly, CFA has now become the preferred means to measure continuous high-resolution $CH_4$ records on ice cores due to the efficient sampling possible in a CFA setup (Schüpbach et al., 2009; Stowasser et al., 2012; Rhodes et al., 2013).





After its first successful application on the GRIP ice core (Fuhrer et al., 1993), the University of Bern CFA system has been
      further developed and extended to other analytes and has been used for a number of major deep ice-core campaigns and many
      shallow cores both in the field in Greenland and Antarctica as well as in the lab. The deep ice cores include, among others, the
      NGRIP (North Greenland Ice Core Project) and NEEM (North Greenland Eemian Ice Drilling) ice-cores of which the data is
      presented here.

Both the NGRIP and the NEEM CFA datasets have been used in a wide variety of studies. They range from dating of the
      NGRIP ice core by annual layer counting to detailed investigations of past climate and its variability. Each of the studies
      highlighted in the following exploits the characteristics of the CFA datasets in different ways. Overall, they can serve as select
      examples demonstrating the range of questions that are possible to address with the datasets released alongside this paper.

      Leveraging the full 1 mm resolution NGRIP data, the deeper section of the GICC05 age scale from 10 to 42 ka b2k was
made by counting annual layers using the seasonal variations of the aerosol concentrations in the ice (Rasmussen et al., 2006;
      Andersen et al., 2006; Svensson et al., 2008). The age scale has subsequently been the foundation of many other ice-core
      chronologies in Greenland through volcanic match points, e.g. for NEEM (Rasmussen et al., 2013) and, most recently for
      EastGRIP (Mojtabavi et al., 2020).

      Together with the dating by annual layer counting, the high resolution CFA aerosol data can, for example, be used to study
episodic aerosol signals. This has been done for the NGRIP and the upper part of NEEM $NH_4^+$ records, where changes in the
      occurrence rates of wildfire plumes reaching the Greenland ice sheet from North America were studied (Fischer et al., 2015;
      Legrand et al., 2016) building on previous work on the $NH_4^+$ seasonality and its temporal change in the GRIP ice core (Fuhrer
      et al., 1996).

      Employing the full range of analytes accessible with CFA and CFA-derived samples from the NEEM ice core, Schüpbach
et al. (2018) investigated changes in aerosol sources and transport back over the entire last glacial period and into the last
      interglacial, the Eemian. The deconvolution of the influences of source strength and transport efficiency changes on the signal
      in the ice showed only a small change in aerosol source strengths in the last interglacial in comparison to today, despite
      significantly increased concentrations in the ice.

      The precise co-registration of the broad range of different aerosol species in the CFA datasets allows for detailed studies of
the temporal relationships between environmental changes in different parts of the Earth system. The CFA data of NGRIP and
      NEEM at annual to multi-annual resolution has, for example, been used to investigate the timing of the rapid warming events
      during the Last Glacial period and at the onset of the Holocene in multiple studies. They have revealed not only the abruptness
      and the decadal scale delays between warming in Greenland, changes in the North Atlantic and Asian climate (Steffensen
      et al., 2008; Erhardt et al., 2019a), but also the diversity of these events (Capron et al., 2021) due to the natural variability in
the climate system.

      Even though parts of the NGRIP and the NEEM datasets have been used in a range of studies, they have not been released
      in full 1 mm resolution presented here (Erhardt et al., 2021). Here, we provide these datasets at high resolution to the ice-core
      and wider climate community. In the following we will give a general description of the NGRIP and NEEM CFA systems,
      the relevant references and some historical context. After that, the data sets will be presented alongside an assessment of the

Earth System
Science
Data

data quality and potential sources of error, followed by a section discussing the depth and temporal resolution of the dataset. Finally, we will provide some summarizing notes to potential users on the caveats of the datasets. An overview of the two multi-proxy datasets is provided in Figure 1, that shows decadal averages of all data over the complete lengths of the records. These 10 yr-averaged datasets on the current age model are provided alongside the 1 mm data as well (Erhardt et al., 2021).

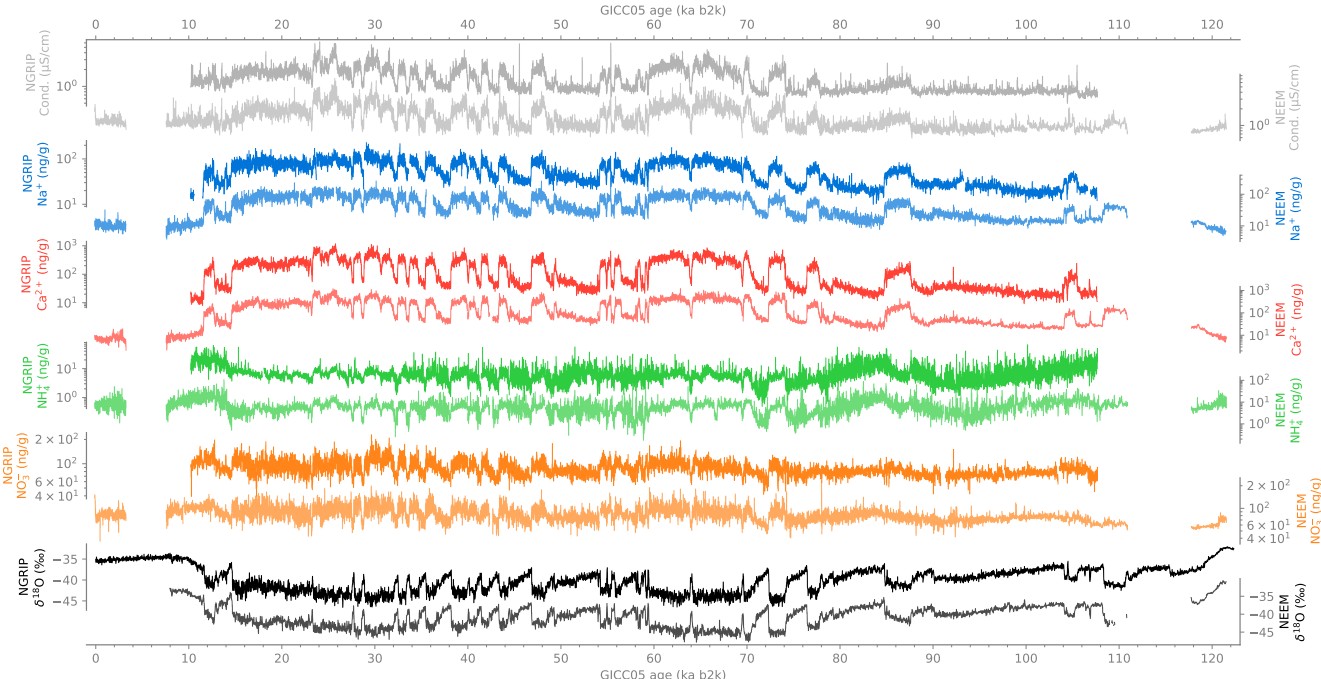

**Figure 1.** All aerosol time series as 10 yr averages on the GICC05 timescale. Data from the NGRIP core are shown in darker colors with scales on the left hand side, data from the NEEM core in lighter colors and scales on the right hand side. The two large gaps in the NEEM data are from the brittle zone between 3.3 and 7.6 ka b2k and due to the over-folding at the bottom of the ice core from 110–114 ka b2k. The two lowermost pannels show the NGRIP and NEEM water $\delta^{18}$O as 20 yr and 30 yr averages (NGRIP project members, 2004; Gkinis et al., 2014, 2021)

.

## 2   Coring locations

The drilling locations of the NGRIP and NEEM ice cores are shown in Figure 2 alongside other deep-drilling sites in Greenland. Both drill sites are located on the divide of the Greenland ice sheet north-west of its summit, the location of the GRIP and GISP2 ice cores.



## 2.1   NGRIP

The NGRIP drill site is located at 75.10°N, 42.32°W and an altitude of 2921 m a.s.l., approximately 315 km NW of the

Greenland Summit. At the site, the mean annual temperature is -32 °C and the mean annual amount of snow accumulation is
0.195 m of ice equivalent (Dahl-Jensen et al., 2002). Between 1996 and 2000, two ice cores were drilled at the NGRIP site,
NGRIP1 and NGRIP2. During the drilling of NGRIP1, the drill got stuck and was lost in 1997 at 1301 m depth, prompting the
drilling of the NGRIP2 core that reached bedrock at 3,085 m depth in 2003 (NGRIP project members, 2004; Dahl-Jensen et al.,
2002). The data presented here is exclusively from the NGRIP2 ice core, covering the depth interval from 1280 m to 2930 m,

the depth reached at the end of the drill season in 2000.

## 2.2   NEEM

The drill site of the NEEM deep ice core is located 350 km NW of NGRIP at 75.45°N, 51.06°W., 2450 m a.s.l.. The site
features annual mean temperatures of -29 °C and a mean annual accumulation of 0.22 m ice equivalent (NEEM community
members, 2013). The NEEM ice core, drilled between 2008 and 2010 reached a total depth of 2544 m below the surface

(Popp et al., 2014; NEEM community members, 2013). The core was continuously measured using CFA methods and the
data presented here cover the complete depth of the core, except for the brittle ice between 650–1178 m depth due to poor ice
quality that prevented the use of CFA.

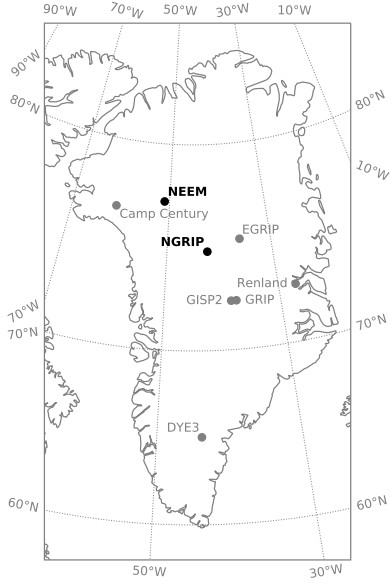

**Figure 2.** Locations of deep Greenland ice cores with the locations of NGRIP and NEEM highlighted in bold. The drill location of NGRIP is
located at 75.10°N, 42.32°W, 2921 m a.s.l., that of NEEM at 75.45°N, 51.06°W., 2450 m a.s.l. (Dahl-Jensen et al., 2002; NEEM community
members, 2013).




## 3 Continuous Flow Analysis

In Continuous Flow Analysis (CFA) ice-core samples are melted and analyzed continuously to obtain high-resolution impurity
concentration profiles along the core. To do so, longitudinal sections of the ice core are cut and placed vertically on a purpose-
built hot-plate to melt along the coreing direction at a slow and controlled speed. This so-called melthead is designed in such a
way that the ice-core sample is efficiently and effectively decontaminated during melting: It is split into an inner and outer area
divided by a small edge, separating the clean melt water from the center of the sample from the possibly contaminated water
stream from the outside of the stick of ice. In this way, the decontamination of the sample is performed without the inner, clean
part of the ice ever coming into contact with the lab environment. Only the melthead comes into contact with sample during
the melting, after that, the sample stream runs through inert PFA or PEEK (Perfluoroalkoxy alkane, Polyether ether ketone)
tubing to the analyzer part of the system.

To be able to measure the small-scale variability of the aerosol concentrations in the ice, continuous detection methods with
short response times are needed. Furthermore, these methods need to be able to measure the low concentrations observed in
the ice. For the GRIP CFA system, Sigg et al. (1994) used spectrophotometric methods, specific for each analyte and modified
for the application in a CFA system to measure the concentrations of dissolved calcium first by absorption and ammonium,
formaldehyde, hydrogen peroxide,and later calcium, by fluorescence detection in the ice at sub-cm resolution. These methods
formed the base for the methods employed for the NGRIP and NEEM CFA records presented here. Figure 3 shows pictures of
the meltheads and the analysis systems used for the NGRIP and NEEM records during measurement campaigns.

### 120 3.1 NGRIP CFA measurements

After the aforementioned very successful use during the GRIP project, the original CFA system of the University of Bern was
further developed for the Antarctic EPICA Dome Concordia (EDC) and Dronning Maud Land (EDML) projects (Röthlisberger
et al., 2000; Fischer et al., 2007; Wolff et al., 2006). During the same time the system was also used for the Greenland NGRIP
ice core (Dahl-Jensen et al., 2002). The NGRIP system and its differences to the GRIP system (Sigg et al., 1994) are described
in detail in Röthlisberger et al. (2000) and will be outlined in the following, alongside the relevant references for the individual
methods. The system was field-deployed at Dome C for multiple Antarctic field seasons from 1997–2003 and in 2000 at the
NGRIP drill site in Greenland.

Overall, the system improvements aimed for the addition of further analytes and increased stability and automation to allow
for efficient and repeatable measurements of up to $35\,\mathrm{m}$ of ice per measurement day. The NGRIP system used mostly the
same analytical methods as the GRIP system, however additional analytes were added: $Na^+$, $NO_3^-$ (EDC/EDML/NGRIP) and
$SO_4^{2-}$ (NGRIP), all based on absorption spectroscopic techniques. Overall, the system was finally able to detect seven species
(calcium, sodium, ammonium, nitrate, sulphate, hydrogen peroxide and formaldehyde) using spectrophotometric methods.
Furthermore, a commercially available conductivity meter was added to the system to measure the electrolytic conductivity of
the meltwater. Additionally, the system supported other instruments such as a laser particle counter and sizer (Ruth et al., 2003,
135 2007).

**Figure 3.** Pictures of the meltheads and analysis systems used for the NGRIP and NEEM ice cores. Panels A and C show the NGRIP melthead and system during the NGRIP field deployment, Panels B and D the melthead and system in use at NEEM. (All pictures by Matthias Bigler)





Based on the previous developments, detection of $NH_4^+$ was performed with the same fluorimetric method described in Sigg et al. (1994) (Genfa and Dasgupta, 1989). The detection of hydrogen peroxide and formaldehyde also used the previously developed methods, however with improved spectrometers (Dasgupta and Hwang, 1985; Dong and Dasgupta, 1987). Differing from the GRIP system, the EDC/NGRIP system used a more sensitive fluorimetric detection of $Ca^{2+}$ (Tsien et al., 1982) instead

of the previously used absorption technique (Kagenow and Jensen, 1983). The main advantages of the fluorimetric detection over the absorption method are higher sensitivity, lower limits of detection and higher resolution due to the much smaller flow cell (Röthlisberger et al., 2000). Detection of dissolved $Na^+$ was achieved using an enzymatic reaction originally developed for flow injection analysis (FIA) (Quiles et al., 1993) and further adapted for the application in the CFA system (Röthlisberger et al., 2000). $NO_3^-$ was detected using a well-established absorption technique (McCormack et al., 1994) based on the reduction

of nitrate to nitrite using copper-coated cadmium reactors, which are custom made for the CFA system (Röthlisberger et al., 2000).

To provide enough water for the additional CFA channels as well as the collection of discrete samples for off-line analysis, the original melthead of the GRIP system was redesigned for a larger sample size. Whereas the GRIP system used an 18 by 18 mm cross section of the ice core with an inner circular melthead area of 11 mm diameter, the NGRIP system uses a 31 by

31 mm cross section with a 20 mm inner diameter, made out of gold plated copper as shown in Figure 3A.

During the only deployment to NGRIP in the 2000 field season, a total of 1525 m of ice (1405 to 2930 m depth of the NGRIP2 ice core) were melted and analyzed using the NGRIP CFA system. Analysis was performed on 1.65 m long segments of ice at meltspeeds of approximately 4 cm min$^{-1}$. No CFA measurements were performed on the shallower part of either of the NGRIP ice cores.

The data from the NGRIP ice core provided here consists of continuous records of dissolved calcium, sodium, ammonium and nitrate as well as the electrolytic melt water conductivity. Even though more parameters were measured during the campaign, namely sulphate, dust particles, formaldehyde and hydrogen peroxide, they suffer from larger analytical uncertainties and are partly of much lower quality and are thus not provided here.

The analyzed depth range of the NGRIP2 ice core covers the age range of 10280 a b2k to 107600 a b2k on the GICC05modelext

age model (Rasmussen et al., 2006; Andersen et al., 2006; Svensson et al., 2008; Wolff et al., 2010).

An example of the NGRIP CFA data is shown in Figure 4: The 100 a section shown depicts the rapid transition from the Younger Dryas cold period into the preboreal Holocene (NGRIP project members, 2004; Walker et al., 2009). Both $Na^+$ and $Ca^{2+}$, show clear seasonal cycles and reveal the rapid nature of the transition in the impurity records, as well as the equally rapid change in annual layer thickness as indicated in Figure 6.

## 3.2 NEEM CFA measurements

Unfortunately, the system previously used for the NGRIP measurements was lost in transport during the return from the 2002/2003 field season at Dome C. Though disruptive, the loss of the system offered a chance for a complete re-design of the Bern CFA setup. The new system is described in detail in Kaufmann et al. (2008), and in the following we will provide an overview of the improvements. During its development phase, the new CFA system was deployed multiple times at the Alfred

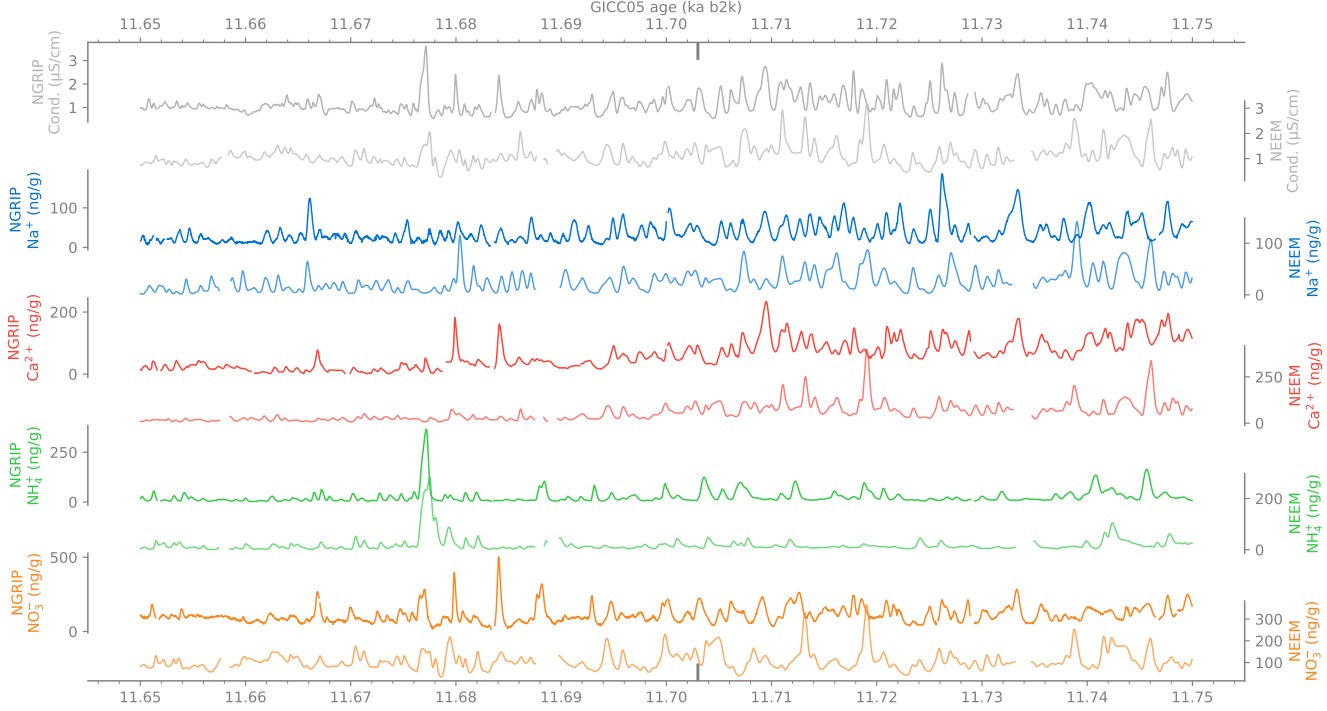

**Figure 4.** Full 1 mm-resolution data sets from both cores over the transition from the Younger Dryas cold period to the Preboreal Holocene as marked by the two vertical markers at 11.703 ka b2k (Rasmussen et al., 2014). Data from the NGRIP core is shown in darker shading with axis on the left side, data from the NEEM core in pale colors with scales on the right hand side. The annual cycles, especially visible in the calcium and sodium records, indicate the rapid increase of annual snowfall over the transition, going hand in hand with large concentration changes. Especially noteworthy is the clear $NH_4^+$-spike visible in both ice-cores, likely a signal from a North American wild-fire.

Wegener Institute for Polar and Marine Research in Bremerhaven, Germany for the analysis of the Antarctic EPICA Dronning Maud Land (EDML) core in 2004 to 2006 and for the Talos Dome Ice Core (TALDICE) in 2006 to 2008 (Fischer et al., 2007; Wegner et al., 2015; Schüpbach et al., 2013). During both of these campaigns a number of innovative ideas where tested that later found their way into the NEEM system. After the campaigns, and some final improvements, the system was deployed over three field seasons at the NEEM deep drilling camp. This system is still in use today with only minor changes.

Although the new system was redesigned and rebuilt from the ground up (Kaufmann et al., 2008) it was based around the same analytical techniques as the EDC/NGRIP system (Sigg et al., 1994; Röthlisberger et al., 2000). However, it featured a more compact, more modular design, aimed at an easier field deployment. For this, the system is split into a melting, a water-distribution and an analysis module that can be packed separately into standard boxes for shipping. Additionally, a large fraction of steps involved in the calibration and analysis runs were further automated to facilitate more efficient and more robust

measurement procedures.



In the custom-built detectors of the new system, the previously used phosphor-coated mercury and cadmium lamps were replaced by light emitting diodes. Furthermore, across all detectors, the same photo multiplier modules were employed that were previously only used in the sodium and nitrate channels. The newly built spectrometers and the further optimized setup of the lines in the individual channels overall improved the response time of the detection methods, especially for $Ca^{2+}$ and $NH_4^+$

as shown in Table 1 and discussed later. To increase the reliability and long-term signal stability, active temperature control was added to each of the analytical sub-systems as well as to the heating/cooling loops used to control some of the analytical reactions (Kaufmann et al., 2008).

The melthead of the NEEM CFA system was also re-designed to more efficiently use the available sample cross section. It uses a square design for 36 by 36 mm ice samples with an inner square area of 26 by 26 mm (Bigler et al., 2011). The new

square design uses more than 50 % of the sample cross section for the clean meltwater stream, whereas the previously used round versions used around 30 %. This more efficient use of the sample cross section and the overall increase of the sample provides more clean melt water per given depth interval. This allows to either produce a larger volume of sample at a given melt speed for additional sampling efforts alongside the CFA as e.g. during the NEEM melting campaign (Schüpbach et al., 2018), or to decrease the melt speed to increase the measurement resolution for specific CFA applications (Bigler et al., 2011;

Bohleber et al., 2018; Svensson et al., 2011).

Additionally, the system features a debubbler designed to be completely airtight towards the ambient air. In this way, the air bubbles trapped in the ice and transported by the meltwater stream can be extracted and used for gas measurements. These measurements include the estimation of the total air content of the ice (Federer et al., 2008) as well as the pioneering measurement of methane concentrations by gas chromatography and laser spectroscopy (Schüpbach et al., 2009; Stowasser

et al., 2012).

The NEEM CFA data set provided here covers a total of 2.2 km of core and was measured in three consecutive field seasons covering the depths of 0–600 m in 2009, 1281-2200 m in 2010 and the brittle ice from 600–1281 m in 2011. Analysis of the NEEM ice core was performed on 3.5 by 3.5 by 110 cm sections of the ice core at melt speeds around 3.5 cm min$^{-1}$. Though efforts were made to preserve the brittle core by letting it relax for a year after drilling and adjusting handling procedures,

a large part of the ice from the brittle zone was compromised by cracks and could not be measured reliably using the CFA methods. This leads to a large gap in the final CFA datasets spanning the depth range of 650–1178 m depth.

Like the NGRIP dataset, the NEEM CFA dataset presented here consists of the concentration records of calcium, sodium, ammonium and nitrate as well as the electrolytic meltwater conductivity. The data cover the time back to 128.6 ka b2k with a gap between 3.3 and 7.6 ka b2k on the GICC05modelext-NEEM-1 age scale (Rasmussen et al., 2013). In the lower part of the

NEEM core, the stratigraphy is not preserved due to folding of the ice. However, the age scale for this section of the core was successfully reconstructed by unfolding its gas records to match their Antarctic counterparts (NEEM community members, 2013). Due to the over-folding, some age intervals occur twice in the record (107–119 ka b2k) or are missing from the vertical section of the ice sheet at this position (110–114 ka b2k). Both data gaps are clearly visible in the 10 yr-resolution aerosol records shown in Figure 1 alongside the corresponding data from the NGRIP core.





## 4   Data quality

Even though the measurement systems used for the two ice cores are quite different, both the NGRIP and NEEM datasets feature similar quality characteristics as listed in Table 1. This is partly due to comparable sample preparation and melting techniques, as well as the identical analytical techniques used. Most of the improvements between the two systems were aimed at ease of use, efficiency and the long-term stability of the system and did not necessarily yield large analytical improvements. Nevertheless, signal response times are overall better on the NEEM system.

The requirement for good core quality is one of the biggest drawbacks of all continuous melting methods, but especially for the very contamination sensitive chemical analysis presented here. Both reliable, stable melting as well as the de-contamination in CFA is virtually impossible to achieve in samples containing any breaks. Even cracks in the ice that are refrozen are often highly contaminated with drillliquid and need to be removed prior to analysis. Furthermore, the dimensions of the CFA sample need to be well constrained, both in straightness and cross-section, to allow for efficient sample use and stable melting conditions. Though this requirement seems trivial, in practice it is not always easy to meet during sample preparation. Especially in sections with lower core quality, such as the brittle zone of deep ice cores, these constraints on the sample quality often lead to large data gaps.

In the following sections we provide an overview of the known sources of errors in the datasets regarding the analytical precision and the depth assignment. We also discuss the processes that determine the actual resolution of the datasets both in terms of depth as well as time.

### 4.1   Analytical precision

A large part of the uncertainty in the concentration measurements is a result of the calibration procedures as detailed for the NEEM data in Gfeller et al. (2014). Due to the need for efficiency during the measurement campaigns, all calibration standards are prepared in two dilution steps, using a calibrated water dispenser (Dispensette, Brand) and micro-liter pipettes (Socorex) which lead to larger errors in the final concentrations of the standards as compared to, e.g., gravimetric preparation or volumetric flasks. Estimating this error by Gaussian error propagation and using weighted regression, average calibration uncertainties for all ions shown here are given alongside typical limits of detection in Gfeller et al. (2014). Relative uncertainties for each of the analytes are typically below $10\%$ for the lower concentration ranges during warm climate periods shown here, and are lower for the higher concentrations during cold periods. Overall, these values are in good agreement with the deviations between repeat measurements for the EDC/NGRIP system (Röthlisberger et al., 2000) as well as for the NEEM system (Kaufmann et al., 2008). Determining the limit of detection as three times the standard deviation of blank measurements of ultra pure water, the typical detection limit for the ion records presented here is $0.1\,\mathrm{ppb}$ (Röthlisberger et al., 2000; Kaufmann et al., 2008; Gfeller et al., 2014).

The accuracy of CFA measurements can also be assessed by comparing CFA results to those from other analysis methods for ice-core impurities, such as ion chromatography and inductively coupled plasma mass spectrometry. Typically, these measurements are either performed on-line (ICP-MS), semi-continuous (Fast-IC or FIC), or off-line on discrete aliquots from





**Table 1.** Limits of detection (LOD), temporal resolution of the detection channels (e-folding time, $\tau$, and 10–90 % time, $t_{10-90}$) and resulting depth resolution (Res.) based on $\tau$ at the melt speeds used to generate the respective data sets. Note, that the resolution does not account for additional signal dispersion during the melting and by the debubbling volume and is only a lower limit.

|  |  | LOD (ppb) | $\tau$ (s) | $t_{10-90}$ | Res. (cm) |
|---|---|---|---|---|---|
| NGRIP[1] | $Ca^{2+}$ | 0.1 | 18 | 40 | 1.2 |
|  | $Na^+$ | 0.1 | 16 | 35 | 1.1 |
|  | $NH_4^+$ | 0.1 | 15 | 33 | 1.0 |
|  | $NO_3^-$ | 0.1 | 16 | 35 | 1.1 |
| NEEM[2] | $Ca^{2+}$ | 0.1 | 7 | 15 | 0.7 |
|  | $Na^+$ | 0.1 | 15 | 33 | 0.9 |
|  | $NH_4^+$ | 0.1 | 10 | 22 | 0.6 |
|  | $NO_3^-$ | 0.1 | 15 | 33 | 0.9 |
|  | Cond. |  | 7 | 15 | 0.5 |

Values from:
[1]Röthlisberger et al. (2000); 10–90 % time converted from e-folding time, depth resolution assumes a melt speed of $4\,\mathrm{cm\,min^{-1}}$, no value provided for conductivity
[2]Kaufmann et al. (2008); e-folding time converted from 10–90 % time, depth resolution assuming a melt speed of $3.5\,\mathrm{cm\,min^{-1}}$

the CFA system (IC, ICP-MS). In an extensive comparison between results from various IC labs and CFA measurements preformed on the Dome C ice core, Littot et al. (2002) show the overall good agreement between IC and CFA measurements with the EDC/NGRIP system. Though there are sometimes slight differences between IC and CFA measurements, these are often due to method differences such as additional contamination risks when discrete samples are handled during IC analysis. They point out, that in the case of ammonium, CFA is preferable to IC measurements because of the increased contamination risk during the IC measurements caused by the environmental prevalence of ammonium and ammonia in (the lab) environment. A systematic comparison of measurement techniques and proxies for mineral dust in polar ice, including CFA and IC $Ca^{2+}$ measurements also indicates an overall good agreement beteen the concentrations obtained by the two methods (Ruth et al., 2008). More recently, Erhardt et al. (2019b) showed excellent agreement between CFA calcium and sodium concentrations and on-line ICP-MS measurements in a direct comparison using the NEEM CFA system on Holocene ice from the EastGRIP ice core. The correlation between CFA and ICP-MS data was 0.96 with low ppb RMS differences between the methods despite the different smoothing by dispersion in the different lines, and the fact that ICP-MS measures elemental concentrations whereas the CFA measures dissolved ions.

### 4.1.1 NEEM Calcium

The NEEM calcium record presented here warrants some additional deliberations: The deep drilling at NEEM used a then new drill liquid based on a mixture of COASOL and ESTISOL 240 (Popp et al., 2014). This mixture has been proven as an



excellent alternative to the hydro-carbon/hydro-fluorocarbon based drill liquids that were previously used in other deep-drilling

projects (Sheldon et al., 2014).

As described in the appendix of Schüpbach et al. (2018), COASOAL has a substantial influence on the determination of $Ca^{2+}$ and likely other bivalent cations. During the drilling and core-handling drill liquid can enter the ice cores through small cracks. These cracks are often re-frozen and very hard to see and can –if not removed– lead to an introduction of small amounts of drill liquid into the CFA system. In the PEEK and PFA tubing of the CFA system, the drill liquid seems to act

as an absorber for calcium ions and leads to large adsorption/desorption features in the CFA $Ca^{2+}$ data. These features are especially pronounced at the beginning of the measurement runs leading to a loss in signal amplitude and a slow increase of the baseline and, if not corrected, also the average concentrations over each run until an adsorption/desorption equilibrium is reached. This effect is illustrated by the example shown in Figure 5. The gray line in Figure 5 shows the slow increase of the Ca concentrations in two consecutive measurement runs.

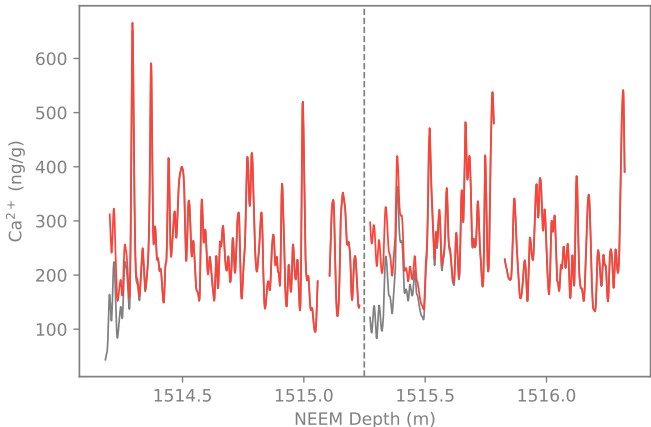

**Figure 5.** Example of two consecutive measurement runs exhibiting the absorption/desorption features caused by the drill-liquid contamination in the CFA system. The two runs are separated by the vertical dashed line. The gray line shows the uncorrected data, the red line is the empirically corrected data (Schüpbach et al., 2018). The correction only affects the beginning of each run. After the adsorption/desorption equilibrium is reached, the two lines overlap.

Schüpbach et al. (2018) used an empirical correction that corrects for this artificial trend by assuming the that the average concentration during the second half of each 110 cm measurement run is representative for the whole run as illustrated in Figure 5 by the red lines. The correction uses an exponential fit (justified by the Langmuir adsorption theory) to the data that is then used to correct the lower concentrations at the beginning of the measurements. This correction has also been applied to the $Ca^{2+}$ data presented here and implies that the variability of the data at the cm-scale should only be interpreted with great

care. The overall very good coherence of the corrected NEEM $Ca^{2+}$ record and its NGRIP counterpart, which is not subject to this drill fluid effect, supports that the longer-term variations in the NEEM $Ca^{2+}$ record are well preserved and of atmospheric origin.



It is worth stressing, that the adsorption/desorption problem very likely not only affects the CFA $Ca^{2+}$ measurements but all bivalent cation measurements performed both on CFA-meltwater and possibly also on dedicated ice aliquots, depending on sample preparation. At present, there seems no easy way of removing the contamination from measurement systems without replacing all tubing, meaning that contamination will accumulate and affect all samples measured after a drill-liquid contamination. Furthermore, because the contamination acts as an absorber, the error introduced into each measurement is dependent in a non-trivial way on the concentrations measured prior to the affected sample, suggesting that corrections might not be easily possible. This is especially relevant in the light of the continued and very successful use of the COASOAL/ESTISOL 240 mixture in other deep drilling campaigns and calls for more careful decontamination and quality-control protocols to mitigate its detrimental influence on the impurity measurements.

## 4.2 Depth assignment

The uncertainty of the depth assignment of the CFA data is the result of a number of different steps from the logging of the ice core to the synchronization of the parallel measurements in the CFA systems. Each of these steps carries its own uncertainty affecting both relative and absolute depth assignment of the individual CFA records. Here we split the steps into three main sources of uncertainty: The between-channel phasing for the different analytes, the measurement-time to sample-position assignment during the melting and finally the absolute depth assignment within the core.

To record the melting process and assign measurement time to a given depth along the sample, both CFA systems use rotary encoders and weights on top of the samples. Because all subsystems have different lengths of tubing and flow rates they all exhibit different delays from the melting of the ice, i.e., the melthead, to the detection in the respective detectors. With the NEEM system, these delays are regularly determined by injecting a short pulse of a multi-component standard solution just down-stream of the debubbler, before the meltwater stream is split up into the different channels. The delays of the resulting peaks in each of the detectors are then used to align the data sets to each other and the melting. During the NGRIP measurements the signal change in the individual channels from the ultra-pure water baseline to the sample was used. The time it takes for the meltwater from the melthead to reach the debubbler is determined using the beginning and end of each measurement run. Together with the encoder data, these delays provide the basis for the depth assignment of the individual datasets within each measurement run.

The relative depth assignment between the different CFA components is only dependent on the identification of the alignment signals and the stability of this alignment over time. For the NGRIP data the alignment is performed at the beginning of each measurement run, for the NEEM data, the relative phasing is determined during each calibration run, every 2–3 measurement runs, or whenever changes to the analytical systems are made. The resulting delays between the channels are stable over long periods of time and multiple runs, barring any changes to the analytical system such as pump-tube or tubing replacements. Typically, the alignment of the peaks can be done precisely to $\pm(1-2)\,\mathrm{s}$ of measurement time, the alignment of the signal onset as done in the NGRIP data is slightly less precise depending on the concentration of the ice. At the usual meltspeeds of $3–4\,\mathrm{cm\,min^{-1}}$, this translates to a relative depth uncertainty between the CFA components on the order of a millimeter to a few mm in the worst case.



The uncertainty of position assignment of the CFA data along the individual 1.1 or 1.65 m ice samples of each run is determined by the accuracy of the rotary encoder and the accuracy of the sample preparation before the measurement. To ensure stable melting and reliable decontamination every break surface in the ice needs to be decontaminated and made co-planar with the melthead. Like the total length of the sample, the length of ice removed around each break is determined following a strict protocol to ensure the highest possible precision. Nevertheless, the typical precision of each of these measurements is on the order of a few mm (typically 1–3 mm) due to, for example, parallax effects and the working conditions in the cold. In comparison to the manual measurement of ice samples with a ruler during sample preparation, the uncertainty of the depth assignment by the encoder is negligible due to the very high resolution of the used encoder (<0.1 mm). To gauge the combined uncertainty, it is possible to compare the estimated length of ice melted for a given run using the encoder to the length of ice determined during sample preparation. Typically, these two estimates agree with deviations on the order of tens of permill of the total length measured, indicating the error to be on the order of a usually a few millimeter to up to a centimeter in rare cases when the sample has a lot of breaks, or whenever errors were made during sample preparation.

Direct assessment of the combined uncertainties is possible using repeated measurements of ice samples that are horizontally next to each other in the core. Röthlisberger et al. (2000) report that the depth assignment agrees within ±1 cm for triplicate measurements using the EDC/NGRIP System. Repeat measurements using the NEEM system indicate a similar order of magnitude for the combined uncertainty of between-channel alignment and sample-position assignment (Hiscock et al., 2013).

Finally, the depth assignment of the individual CFA samples to an absolute depth along the ice core is determined by the logging and cutting procedures during the processing of the ice core. The logging method and a detailed error analysis is presented in Hvidberg et al. (2002) for the NGRIP ice core and the same method has been used on the NEEM core. During processing the ice core is cut into equal length sub-samples, that determine the absolute depth assignment within the core at 0.55 cm intervals, referred to as bags. The top-most edge of these sub-samples is used as a datum for the relative depth assignment within each bag, calculating the absolute depth from the continuously numbered bags. Markers for the cuts splitting the individual bags are placed following a strict protocol that avoids any accumulation of errors, nevertheless cutting of the sub-samples introduces slight uncertainties, for example, due to the blade thicknesses of the saws or parallax errors. However, this source of uncertainty applies to all measurements on the bags and any error will effect the measurements in the same or very similar way.

In summary, this means that the assignment of the measurements within the sample intervals, typically within 0.55, 1.1 or 1.65 m, determines the overall accuracy of the depth assignment between CFA and other measurements and can be assumed to be around ±1 cm as corroborated by repeat measurements (Röthlisberger et al., 2000; Hiscock et al., 2013). However, within each of the CFA data sets is the relative depth assignment between the different impurities is much more accurate to about a millimeter.

## 4.3 Depth and temporal resolution

Even though CFA employs continuous measurement techniques and is made available at 1 mm sampling interval, the real resolution of the data is lower. Two processes affect the actual resolution of the CFA data: Signal dispersion due to mixing





at the melthead, the debubbler, in the tubes, the reaction columns (sodium and nitrate) and in the flow-through cuvettes of the spectrometers. This process acts as low-pass filter, smoothing the signals preserved in the ice and leading to a significant reduction in the depth resolution of the data. The resulting response time of the detectors can be determined experimentally using controlled injections of standard solutions into the water streams. As this processes have a constant smoothing lenght

in terms of measurement time, the resulting depth resolution of the data sets is a function of the melt speed with lower melt speeds leading to higher resolution.

The practical implications of the smoothing processes for the resolution of the CFA data were already investigated in detail in the initial CFA publication of Sigg et al. (1994). To asses the relative contributions of the melting system and the analyzers to the signal dispersion they employ step concentration changes introduced both at the melthead and directly upstream of the

analyzers and a time-domain restoration filter discussed below. These experiments indicate that for absorption spectroscopic techniques, the dispersion in the large absorption measurement cells of the analyzers and the reaction columns dominates the overall smoothing of the signals. In the case of fluorescence methods, where very small measurement cells can be used, the total dispersion is dominated by the melting system (Sigg et al., 1994). Furthermore, longer tubing lengths, as needed, for example, for the chemical reactions to take place in some of the methods, can significantly contribute to the dispersion in the

individual channels, even if similar spectrometers are employed.

The resolution of the different analysis channels are listed in Table 1 alongside the resulting depth resolutions at the respective meltspeeds. The values listed in the table and discussed below are only a lower limit of the usable resolution, as the signal dispersion by the melting and debubbling volume is not accounted for in these numbers and adds additional smoothing. For the NGRIP/EDC system the resolution of the analytical subsystems was determined as the e-folding time of the signal after a

step-change in concentration (Röthlisberger et al., 2008). These experiments indicate a smoothing of approximately 15–18 s for the ion concentrations presented here, translating into a depth resolution of approximately 1 cm at the employed meltspeed of 3–4 cm min$^{-1}$. Kaufmann et al. (2008) report the temporal resolution of the NEEM CFA system as determined by the 10–90 % rise time as 14 s for conductivity and calcium, 20 s for ammonium and 32 s for sodium and nitrate. Converted to an e-folding time these values are shown in Table 1. They indicate a slight improvement of the analyzer resolution in the NEEM system in

comparison to the NGRIP system likely due to the improved spectrometers and the further optimization of the tubing/mixing setups of the individual channels. At the typical meltspeeds used for the NEEM campaign around 3.5 cm min$^{-1}$, the e-folding times correspond to a depth resolution of just below 1 cm for the channels with the highest smoothing. It is also worth noting, that the smoothing not only decreases the resolution of the data but also adds considerable amounts of auto-correlation to the signals that needs to be taken into account when interpreting the data at cm-scale and sub-cm-scale resolution.

Considering the smoothing as a first-order low pass filter, a smoothing length of 1 cm indicates, that the amplitude of signals with wavelengths below approximately 6 cm is already reduced to at least half its original value. This is especially important in the context of the seasonal variability of the aerosol concentrations that is used for annual layer counting. As the annual layers become thinner with increasing depth due to glacier flow, their signal can be partially or completely smoothed out by the smoothing of the CFA system. Because this loss of signal happens gradually down-core it is easy to misinterpret the remaining

variability as seasonal signals, leading to increased uncertainty of the annual layer count or even systematic under-counting. To





recover the loss in high-frequency variability due to the smoothing, different restoration approaches have been proposed over the years. All of the restoration filter approaches base on estimates of the mixing lengths for the different components. The first of these approaches, proposed already in the first CFA method publication (Sigg et al., 1994), is based on a time-domain finite impulse response (FIR) filter. The only free parameters of the FIR restoration filter are the mixing lengths that were determined

empirically using the step-responses from standard measurements. The second proposed response uses a frequency-domain Wiener filter similar to the filters used to restore smoothed water-isotope signals from ice cores via deconvolution (Rasmussen et al., 2005). In this approach the spectral characteristics of the CFA data are used to determine the noise floor due to instrument noise as well as the smoothing filter to construct an optimal filter that restores some of the smoothed-out variability without amplifying the intrinsic measurement noise too much. Fundamentally, both of these approaches are limited by the measurement

noise in the data series that limits the signal/noise ratio in the at higher frequencies. Neither of these methods are used routinely in the data processing and were not applied to the data presented here to avoid the amplification of measurement noise and the resulting introduction of additional variability into the data series.

Finally, for a given depth resolution of the CFA records, the temporal resolution of the CFA data is determined by the age/depth relationship of the respective ice cores. As mentioned before, due to glacial flow, the annual layer thickness $\lambda$

decreases with increasing depths. This results in a reduction of the temporal resolution of the CFA data presented here. Figure 6 shows the annual layer thickness for the NGRIP and NEEM ice cores both as functions of depth and age. For both cores, the CFA data with its effective cm-resolution as indicated by the horizontal dashed line in the figure resolves annual layers throughout the Holocene and the later parts of the last glacial period. However, due to the much lower accumulation rates in the glacial and the increased thinning with depth, annual layers are not resolved in the lower parts of the core, especially in the

colder periods of the last glacial. At NGRIP, stronger basal melting than at NEEM leads to less thinning with depth, making the annual layers resolvable back to approximately 60 ka b2k. Figure 4 illustrates the presence of the pronounced seasonal variations especially in Calcium and Sodium, as preserved over the transition from the Younger Dryas into the preboreal Holocene in both the NGRIP and NEEM ice cores.

That being said, it is worth noting that even if the records presented here technically have a high enough resolution to resolve

the seasonal variability on the records, that they are not necessarily interpretable at annual or sub-annual resolution in terms of aerosol deposition onto the ice sheet. The preserved signal in the ice-core record is affected by a range of processes that lower the representativness of the records at high temporal resolution. The known processes range from post-deposition loss of $NO_3^-$ depending on the burial rate (e.g. Fischer et al., 1998; Fibiger et al., 2013) and intermittency of precipitation and wind re-working of snow at the surface (e.g. Fisher et al., 1985; Casado et al., 2020) affecting the records at the centimeter-scale to

the relocation of chemical impurities in the ice-matrix by re-crystallization and anomalous diffusion (e.g. Faria et al., 2010; Ng, 2021) affecting the records at the millimeter to sub-millimeter scale. In summary at high spatial and temporal resolution, the records are strongly affected by archive specific processes that can alter the climatic information substantially or even obliterate it completely.

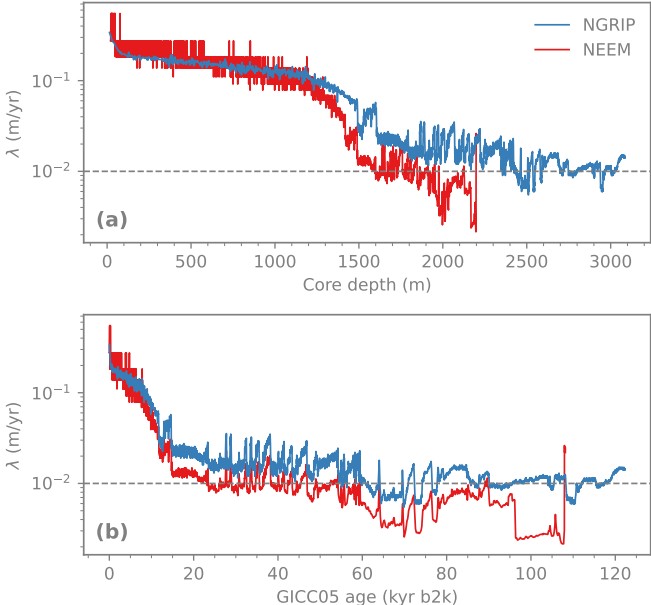

**Figure 6.** Annual layer thickness of the NGRIP and NEEM GICC05modelext age scales as a function of depth (a) and age (b). The horizontal dashed line indicates 1 cm, the effective resolution of the data presented here.

## 5   Conclusions

Here we present multi-proxy records from two deep Greenland ice cores at high resolution. Each of the two records include the concentrations of the major ionic aerosol species, calcium, sodium, ammonium and nitrate as well as the electrolytic melt water conductivity at 1 mm resolution. Both ice-core records cover the entire last glacial period and the transition into the Holocene. With the unprecedented high resolution of of the datasets, they can be used to study northern-hemispheric environmental changes at great temporal detail and have been the foundation of a multitude of ice-core studies.

Nevertheless, summarizing from the considerations above we would like to emphasize the following points for potential users of the datasets provided with this paper:

1. Even though the datasets have been extensively quality checked and cleaned from analytical problems (contamination, air bubbles or instability see e.g. Röthlisberger et al. (2000); Kaufmann et al. (2008)), they are not guaranteed to be free from spurious signals.

2. The depth assignment of the data to a given position along the core is likely on average accurate to better than 1 cm.

3. The relative depth assignment between the analytes within each core is precise on the order of 1 mm.

4. Analytical uncertainties are typically less than 10 % of the stated values and are lower for time intervals with elevated concentrations such as $Ca^{2+}$ and $Na^+$ in glacial times.



5. Due to the signal smoothing during measurement, the usable depth resolution of the datasets is approximately $1\,\mathrm{cm}$. Due to thinning of the annual layers with depth, this resolution translates to a temporal resolution in the range of sub-annual to multi-annual. Special attention has to be paid to the auto correlation of the high-resolution data introduced by the measurement techniques when applying time series analytical techniques. Furthermore, we note, that the high-resolution datasets are strongly affected by archive specific processes which can alter the climatic signals.

6. The $Ca^{2+}$ record from the NEEM core is affected by an interaction with the drill liquid. The record provided here has been empirically corrected. We believe that this correction provides good long-term averages for the calcium concentrations. Nonetheless, special care must be taken when interpreting the record at the centimeter scale.

7. Finally we would like to stress, that any in-detail interpretation of individual short-term signals should always be cross-checked and confirmed at least within the multi-parameter datasets, if not with additional measurements, or data from nearby ice cores.

To avoid some of these pitfalls and shortcomings of these datasets we would like to invite the future users of these data to actively involve CFA or ice-core specialists in their investigations.

## 6  Data availability

NGRIP and NEEM CFA datasets (Erhardt et al., 2021):

– NGRIP CFA data at $1\,\mathrm{mm}$ resolution is available on PANGAEA (https://doi.pangaea.de/10.1594/PANGAEA.935818).

– NEEM CFA data at $1\,\mathrm{mm}$ resolution is available on PANGAEA (https://doi.pangaea.de/10.1594/PANGAEA.935837).

– NGRIP CFA data at $10\,\mathrm{yr}$ resolution is available on PANGAEA (https://doi.pangaea.de/10.1594/PANGAEA.935814).

– NEEM CFA data at $10\,\mathrm{yr}$ resolution, is available in the online supplement of Schüpbach et al. (2018) for calcium, sodium, ammonium and nitrate, and, also including the electrolytic meltwater conductivity on PANGAEA (https://doi.pangaea. de/10.1594/PANGAEA.935831).

Additional datasets used for the figures:

– GICC05modelext for NGRIP is available on https://www.iceandclimate.nbi.ku.dk/data/2010-11-19_GICC05modelext_ for_NGRIP.txt.

– GICC05modelext for NEEM is available on https://www.iceandclimate.nbi.ku.dk/data/2013-12-05GICC05modelext-NEEM-1. xls.

– The reconstructed agescale for the lower part of the NEEM core is available in the online supplement of NEEM community members (2013).



– The NEEM high-resolution water isotope data from Gkinis et al. (2021) is available on PANGEA (https://doi.org/10. 1594/PANGAEA.925552).

*Author contributions.* Both the NEEM and the NGRIP CFA campaigns were team efforts. MB, RR, MLSA, KGA, MEH, UR, BT, and RM
participated in the NGRIP measurements; MB, HF, SS, GG, DL, OS, UF, HAK, PTV, MEH, KGA, TK, AW, BT, KS, NA, ERT, AKB,
LRF and RM in the NEEM measurements both were run under the leadership of the Bern Group which was also responsible for the data
processing and production of the final datasets. The manuscript and figures were prepared by TE with the input of all authors.

*Competing interests.* The authors declare no competing interests.

*Acknowledgements.* The authors also gratefully acknowledge the contributions of the countless people that facilitated and took part in both
the ice-core drilling and processing during the NGRIP and NEEM field campaigns. Furthermore the authors would like to express their
gratitude to Kerstin Schmidt who participated in the measurement campaigns. The Division for Climate and Environmental Physics, Physics
Institute, University of Bern acknowledges the long-term financial support of ice-core research by the Swiss National Science Foundation
(SNSF) under the project numbers 172506, 137635, 147174, 105523, 119612, 159563, 57053 and 63333 as well as by the Oeschger Center
for Climate Change Research. NGRIP is directed and organized by the Department of Geophysics at the Niels Bohr Institute for Astronomy,
Physics and Geophysics, University of Copenhagen. It is supported by funding agencies in Denmark (SNF), Belgium (FNRS-CFB), France
(IPEV and INSU/CNRS), Germany (AWI), Iceland (RannIs), Japan (MEXT), Sweden (SPRS), Switzerland (SNF) and the USA (NSF, Office
of Polar Programs). NEEM is directed and organized by the Centre of Ice and Climate at the Niels Bohr Institute and US NSF, Office of Polar
Programs. It is supported by funding agencies and institutions in Belgium (FNRS-CFB and FWO), Canada (NRCan/GSC), China (CAS),
Denmark (FIST), France (IPEV, CNRS/INSU, CEA and ANR), Germany (AWI), Iceland (RannIs), Japan (NIPR), South Korea (KOPRI),
The Netherlands (NWO/ ALW), Sweden (VR), Switzerland (SNF), the United Kingdom (NERC) and the USA (US NSF, Office of Polar
Programs) and the EU Seventh Framework programmes Past4Future and WaterundertheIce.



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
