# Peer review of "High resolution aerosol concentration data from the Greenland NorthGRIP and NEEM deep ice cores"

_Earth System Science Data, 2021_

## Referee Comment (RC1)

Review of Erhardt et al., 2021, "High resolution aerosol concentration data from the Greenland NorthGrip and NEEM deep ice cores"

For Earth System Science Data

15 December 2021

The authors present the full high resolution (1 mm) datasets from the NorthGRIP and NEEM ice cores, and include a history and perspective on the development of Continuous Flow Analysis systems that led to the datasets. I appreciated the level of detail provided on the analytical systems, the assessment of data quality, and the detailed description of the effective data resolution. These are critical data for understanding the Earth system and Arctic in particular, so I am glad to see this information published in one place, and I feel this makes the article itself appropriate to support the publication of the dataset. The dataset is certainly unique, useful and complete, and of high quality; this paper will be a useful resource for anyone using these datasets in the future. The paper is well written; I only came across two small typos. I recommend that this paper be published essentially as is in Earth System Science Data.

Minor comments:
line 20 - "preserved in the ice as *a* wide range..."
line 35 - "build" should be built

---

## Referee Comment (RC2)

Review of Erhardt et al ESSD manuscript by Bess Koffman

This data description paper presents full-resolution aerosol data from the NGRIP and NEEM ice cores. Although these data have contributed to multiple high-impact papers over the past decades, the full-resolution data have not yet been published. The paper presents the datasets, along with a historical overview of melting and analysis methods used by the Bern ice core lab and associated groups. The full-resolution datasets are likely to be useful for future studies, and I support their publication.

Overall, the paper is straightforward and worthy of publication. I have a few minor suggestions for the authors to consider in the revised version (please see below).

The CFA dataset links work; however, the datasets themselves appear to be under moratorium until March 2022, so I am not able to assess them. The timescale links worked successfully to access text and/or Excel files.

Minor points:

Line 35: "have been built" not "build"

Figure 1: I suggest making it bigger, as the text is barely readable in the current format.

Line 106 and Figure 3: Please add an overhead view of the melthead itself to the figure, rather than having two views of melting with ice sticks. It would be more useful for people reading the paper to see the actual configuration of the top of the melthead.

Lines 121-127: Paragraph could use re-organizing.

Line 129: Please clarify the length of day here – is this 8,10, 24 hours?

Lines 159-160: This is an odd one-sentence paragraph. Please revise.

Line 196: It would be helpful to readers to have more information about the debubbler. Can you add a few more sentences describing how it is built and how it works?

Line 222: should be "analyses" plural

Line 224: "drillliquid" should be two words

Line 275: "the that the" at end of line. Please revise.

Line 320: What exactly is the "strict protocol" used? It would be helpful to readers to have a bit more detail, so they can learn from your well-established approaches.

Line 339: Similar comment to above. A bit more information would be helpful.

Line 352 and related sentences: Dispersion not only affects the resolution of the record, but also the time at which signals arrive (i.e, peak lead phenomenon), with implications for signal deconvolution. A more thorough discussion of dispersion, citing the study of Breton et al., 2012, would be appropriate here.

Line 423: "of of"

Section 6 on data availability:
Just a note that the links seem to work, but the pages that open do not seem unique to each ice core. For instance, the NEEM links go to pages that list "NGRIP" as the project, even if the NEEM data are also linked. I found this confusing.

References cited:
Breton, D.J., Koffman, B.G., Kurbatov, A.V., Kreutz, K.J., Hamilton, G.S., 2012. Quantifying signal dispersion in a hybrid ice core melting system. Environ. Sci. Technol. 46, 11922-11928.

---

## Author Response (AR1)

**Response to Review Comment 1**

*The authors present the full high resolution (1 mm) datasets from the NorthGRIP and NEEM ice cores, and include a history and perspective on the development of Continuous Flow Analysis systems that led to the datasets. I appreciated the level of detail provided on the analytical systems, the assessment of data quality, and the detailed description of the effective data resolution. These are critical data for understanding the Earth system and Arctic in particular, so I am glad to see this information published in one place, and I feel this makes the article itself appropriate to support the publication of the dataset. The dataset is certainly unique, useful and complete, and of high quality; this paper will be a useful resource for anyone using these datasets in the future. The paper is well written; I only came across two small typos. I recommend that this paper be published essentially as is in Earth System Science Data.*

Thank you for your kind words and for sharing our enthusiasm for the data sets.

**Minor Comments**

*line 20 - "preserved in the ice as a wide range..."*

fixed

*line 35 - "build" should be built*

fixed

**General comments**

**Response to Review Comment 2 by Bess Koffman**

*This data description paper presents full-resolution aerosol data from the NGRIP and NEEM ice cores. Although these data have contributed to multiple high-impact papers over the past decades, the full-resolution data have not yet been published. The paper presents the datasets, along with a historical overview of melting and analysis methods used by the Bern ice core lab and associated groups. The full-resolution datasets are likely to be useful for future studies, and I support their publication.*

*Overall, the paper is straightforward and worthy of publication. I have a few minor suggestions for the authors to consider in the revised version (please see below).*

*The CFA dataset links work; however, the datasets themselves appear to be under moratorium until March 2022, so I am not able to assess them. The timescale links worked successfully to access text and/or Excel files.*

Thank you for your review and suggestions which we have incorporated into the manuscript. Private access links where provided to the editorial team with the submission. We were hoping these would be passed along to the reviewers, sorry. The moratorium will be lifted for the final version of the paper.

**Minor Points**

*Line 35: "have been built" not "build"*

fixed.

*Figure 1: I suggest making it bigger, as the text is barely readable in the current format.*

We slightly increased both the size of Figure 1 as well as the font size of the tick labels to increase the readability.

*Line 106 and Figure 3: Please add an overhead view of the melthead itself to the figure, rather than having two views of melting with ice sticks. It would be more useful for people reading the paper to see the actual configuration of the top of the melthead.*

Both Sigg et al. (1994) and Bigler et al. (2011) provide technical drawings of the respective melthead designs. To highlight that and guide the reader towards these we added the following two sentences to the respective paragraphs as well as to the caption of Figure 3:

"Caption Figure 3: Drawings of the melthead designs are provided in Sigg et al. (1994) for NGRIP and in Bigler et al. (2011) for NEEM, respectively."

"NGRIP paragraph: A schematic drawing of the melthead is provided in Sigg et al. (1994)."

"NEEM paragraph: It uses a square design for 36 by 36 mm ice samples with an inner square area of 26 by 26 mm. An overhead view as well as a technical drawing including dimensions of the melthead are provided in Bigler et al. (2011)."

*Lines 121-127: Paragraph could use re-organizing.*

We slightly reworded the paragraph and hope that it is now easier to follow.

"After the aforementioned very successful use during the GRIP project, the original CFA system of the University of Bern was further developed for the Antarctic EPICA Dome Concordia (EDC) and Dronning Maud Land (EDML) projects (Fischer et al., 2007; Röthlisberger et al., 2000; Wolff et al., 2006) as well as the Greenland NGRIP ice core (Dahl-Jensen et al., 2002). The EDC/NGRIP system and its differences to the GRIP system (Sigg et al., 1994) are described in detail in Röthlisberger et al. (2000) and will be outlined in the following, alongside the relevant references for the individual methods. The system was field-deployed at Dome C for multiple Antarctic field seasons from 1997–2003 and in 2000 at the NGRIP drill site in Greenland."

*Line 129: Please clarify the length of day here – is this 8,10, 24 hours?*

This refers to a 24h day, changed accordingly to

"...up to 35 m of ice per 24 h measurement day."

*Lines 159-160: This is an odd one-sentence paragraph. Please revise.*

Merged with previous paragraph(s).

*Line 196: It would be helpful to readers to have more information about the debubbler. Can you add a few more sentences describing how it is built and how it works?*

The debubbler and gas extraction is described in detail in Schüpbach et al. (2009). To highlight this more clearly we change the respective sentence as follows:

"Additionally, the system features a debubbler designed to be completely airtight towards the ambient air as detailed in (Schüpbach et al., 2009). "

*Line 222: should be "analyses" plural*

fixed.

*Line 224: "drillliquid" should be two words*

fixed.

*Line 275: "the that the" at end of line. Please revise.*

fixed.

*Line 320: What exactly is the "strict protocol" used? It would be helpful to readers to have a bit more detail, so they can learn from your well-established approaches.*

Agreed, we added a description of the protocol to the respective paragraph

"For a stable melting and a reliable decontamination every break surface in the ice needs to be decontaminated and made co-planar with the melthead. During the decontamination procedure, the length of the ice before and after decontamination, the break positions as well as the amount of ice removed around each break are recorded using custom fixture and a strict protocol to ensure high precision and repeatability: Initially, the length of the sample before decontamination is recorded after all breaks in the ice have been precisely matched. Prior to decontamination of all break surfaces, each piece is marked and the location of the mark is recorded. After decontamination, the positions of the breaks are recorded from the top of the ice piece. The amount of core-depth removed at each break is then recorded using the marks on each piece as a reference. "

*Line 339: Similar comment to above. A bit more information would be helpful.*

The exact protocol is provided in Hvidberg et al. (2002) and do not want to repeat it in detail here, however to highlight this resource more clearly we changed the sentence as follows:

"The logging method and a detailed error analysis is presented in Hvidberg et al. (2002) for the NGRIP ice core and the same methods have been used on the NEEM core. Here we only provide a brief overview of the procedure."

*Line 352 and related sentences: Dispersion not only affects the resolution of the record, but also the time at which signals arrive (i.e, peak lead phenomenon), with implications for signal deconvolution. A more thorough discussion of dispersion, citing the study of Breton et al., 2012, would be appropriate here.*

We have added the following to the section discussing the signal dispersion.

" As a secondary effect, dispersion of the signals in the CFA system can introduce a small lead of the signal peaks when compared to ideal (plug-flow) conditions (Breton et al., 2012). In their idealized experiments the measurement peaks resulting from delta-shaped, i.e. instantaneous, injection peaks showed that the peak signal arrives earlier than predicted by plug-flow conditions. The resulting peaks where skewed towards earlier times with maximum values being detected before the value of the peak. In their setup this peak-lead phenomenon amounts to an approximate lead by $\sim 5\,\mathrm{mm}$ for delta-shaped peaks between maximum and average values. Though the peak-lead phenomenon was not investigated or quantified for either of the generations of the Bern CFA system shown here, it is worth noting that it likely also affects the data presented here, leading to a small, systematic displacement of the signals towards lower depths on the order of millimeters. However, the amount if peak-lead provided by Breton et al. (2012) is not transferable to other CFA systems and is likely lower for real-world signals. Breton et al. (2012) also pointed out that the peak-lead phenomenon will be different for different channels of a CFA system, due to the difference in their mixing characteristics. In the data here, this differential peak-lead effect is elevated by the alignment of peaks introduced downstream of the debubbler as described above. Furthermore, as pointed out by Sigg et al. (1994), a large part of the total signal dispersion is shared between the channels as it is determined by the melting system. "

*Line 423: "of of"*

fixed.

*Section 6 on data availability: Just a note that the links seem to work, but the pages that open do not seem unique to each ice core. For instance, the NEEM links go to pages that list "NGRIP" as the project, even if the NEEM data are also linked. I found this confusing.*

Thanks for checking this, we have resolved this meta data issue together with the PANGEA team.

**References**

Bigler, M., A. Svensson, E. Kettner, P. Vallelonga, M. E. Nielsen, and J. P. Steffensen (2011). "Optimization of High-Resolution Continuous Flow Analysis for Transient Climate Signals in Ice Cores". In: *Environmental Science & Technology* 45, pp. 4483–4489.

Breton, D. J., B. G. Koffman, A. V. Kurbatov, K. J. Kreutz, and G. S. Hamilton (2012). "Quantifying Signal Dispersion in a Hybrid Ice Core Melting System". In: *Environmental Science & Technology* 46, pp. 11922–11928.

Dahl-Jensen, D., N. S. Gundestrup, H. Miller, O. Watanabe, S. J. Johnsen, J. P. Steffensen, H. B. Clausen, A. Svensson, and L. B. Larsen (2002). "The NorthGRIP deep drilling programme". In: *Annals of Glaciology* 35, pp. 1–4.

Fischer, H., F. Fundel, M. de Angelis, U. Federer, M. Bigler, U. Ruth, B. Twarloh, A. Wegner, R. Udisti, S. Becagli, E. Castellano, A. Morganti, M. Severi, E. W. Wolff, G. C. Littot, R. Röthlisberger, R. Mulvaney, M. A. Hutterli, P. R. Kaufmann, F. Lambert, M. E. Hansson, U. Jonsell, C. F. Boutron, M.-L. Siggaard-Andersen, J. P. Steffensen, C. Barbante, V. Gaspari, P. Gabrielli, and D. Wagenbach (2007). "Reconstruction of millennial changes in dust emission, transport and regional sea ice coverage using the deep EPICA ice cores from the Atlantic and Indian Ocean sector of Antarctica". In: *Earth Planet. Sci. Lett.* 260, pp. 340–354.

Hvidberg, C. S., J. P. Steffensen, H. B. Clausen, H. Shoji, and J. Kipfstuhl (2002). "The NorthGRIP ice-core logging procedure: description and evaluation". In: *Annals of Glaciology* 35, pp. 5–8.

Röthlisberger, R., M. Bigler, M. A. Hutterli, S. Sommer, B. Stauffer, H. G. Junghans, D. Wagenbach, B. Staufer, H. G. Junghans, and D. Wagenbach (2000). "Technique for continuous high-resolution analysis of trace substances in firn and ice cores". In: *Environmental Science & Technology* 34, pp. 338–342.

Schüpbach, S., U. Federer, T. Blunier, P. R. Kaufmann, M. A. Hutterli, D. Buiron, H. Fischer, and T. F. Stocker (2009). "A New Method for High-Resolution Methane Measurements on Polar Ice Cores Using Continuous Flow Analysis". In: *Environmental Science & Technology* 43, pp. 5371–5376.

Sigg, A., K. Fuhrer, M. Anklin, T. Staffelbach, and D. Zurmühle (1994). "A continuous analysis technique for trace species in ice cores". In: *Environmental Science & Technology* 28, pp. 204–209.

Wolff, E. W., H. Fischer, F. Fundel, U. Ruth, B. Twarloh, G. C. Littot, R. Mulvaney, R. Röthlisberger, M. de Angelis, C. F. Boutron, M. Hansson, U. Jonsell, M. A. Hutterli, F. Lambert, P. R. Kaufmann, B. Stauffer, T. F. Stocker, J. P. Steffensen, M. Bigler, M.-L. Siggaard-Andersen, R. Udisti, S. Becagli, E. Castellano, M. Severi, D. Wagenbach, C. Barbante, P. Gabrielli, and V. Gaspari (2006). "Southern Ocean sea-ice extent, productivity and iron flux over the past eight glacial cycles". In: *Nature* 440, pp. 491–496.

---

## Author Response (AR2)

Dear Kirsten,

The datasets are now freely available and the DOIs are registered. The DOIs for the "child"-datasets are still in progress and will hopefully be live soon.

I have updated the links in the manuscript accordingly and have also added the reference to the abstract as per the ESSD policies.

Sincerely
Tobias

---

## Author Response (AR3)

Dear Kirsten,

Good catch, thank you!. I've fixed the typo in the abstract accordingly.

Sincerely
Tobias